# Predictive Factors for Bone Cement Displacement following Percutaneous Vertebral Augmentation in Kümmell’s Disease

**DOI:** 10.3390/jcm11247479

**Published:** 2022-12-16

**Authors:** Xiangcheng Gao, Jinpeng Du, Yongyuan Zhang, Yining Gong, Bo Zhang, Zechao Qu, Dingjun Hao, Baorong He, Liang Yan

**Affiliations:** 1Xi’an Honghui Hospital, Xi’an Jiaotong University, Xi’an 710049, China; 2Medical College, Yan’an University, Yan’an 716000, China

**Keywords:** osteoporosis, Kümmell’s disease, postoperative complications, risk factors, percutaneous vertebral augmentation

## Abstract

Objective: To investigate the independent influencing factors of bone cement displacement following percutaneous vertebral augmentation (PVA) in patients with stage I and stage II Kümmell’s disease. Methods: We retrospectively reviewed the records of 824 patients with stage Ⅰ and stage Ⅱ Kümmell’s disease treated with percutaneous vertebroplasty (PVP) or percutaneous vertebroplasty (PKP) from January 2016 to June 2022. Patients were divided into the postoperative bone cement displacement group (*n* = 150) and the bone cement non-displacement group (*n* = 674) according to the radiographic inspection results. The following data were collected: age, gender, body mass index (BMI), underlying disease, bone mineral density (BMD), involved vertebral segment, Kümmell’s disease staging, anterior height, local Cobb angle, the integrity of anterior vertebral cortex, the integrity of endplate in surgical vertebrae, surgical method, surgical approach, the volume of cement, distribution of cement, the viscosity of cement, cement leakage, and postoperative anti-osteoporosis treatment. Binary logistic regression analysis was performed to determine the independent influencing factors of bone cement displacement. The discrimination ability was evaluated using the area under the curve (AUC) of the receiver operating characteristic (ROC). Results: The results of logistic regression analysis revealed that thoracolumbar junction (odds ratio (OR) = 3.23, 95% confidence interval (CI) 2.12–4.50, *p* = 0.011), Kümmell’s disease staging (OR = 2.23, 95% CI 1.81–3.41, *p* < 0.001), anterior cortex defect (OR = 5.34, 95% CI 3.53–7.21, *p* < 0.001), vertebral endplates defect (OR = 0.54, 95% CI 0.35–0.71, *p* < 0.001), cement distribution (OR = 2.86, 95% CI 2.03–3.52, *p* = 0.002), cement leakage (OR = 4.59, 95% CI 3.85–5.72, *p* < 0.001), restoration of local Cobb angle (OR = 3.17, 95% CI 2.40–5.73, *p* = 0.024), and postoperative anti-osteoporosis treatment (OR = 0.48, 95% CI 0.18–0.72, *p* = 0.025) were independently associated with the bone cement displacement. The results of the ROC curve analysis showed that the AUC was 0.816 (95% CI 0.747–0.885), the sensitivity was 0.717, and the specificity was 0.793. Conclusion: Thoracolumbar fracture, stage Ⅱ Kümmell’s disease, anterior cortex defect, uneven cement distribution, cement leakage, and high restoration of the local Cobb angle were risk factors for cement displacement after PVA in Kümmell’s disease, while vertebral endplates defect and postoperative anti-osteoporosis treatment are protective factors.

## 1. Introduction

Kümmell’s disease, first described by Herman Kümmell in 1891 [1], was an uncommon and complicated spinal condition. In the diagnosis, avascular osteonecrosis of the vertebral body is described as a delayed posttraumatic vertebral collapse, often occurring as a consequence of osteoporotic vertebral compression fractures [2]. The prevalence of Kümmell’s disease in the elderly ranges between 7% and 37%, according to the current literature [3]. With the advent of an aging society, Kümmell’s disease is not uncommon and remains a challenging clinical problem.

Currently, the treatment of Kümmell’s disease is controversial. Conservative management strategies such as bed rest, analgesics, and wearing braces can only relieve part of the pain, and the therapeutic effect is often poor [3]. Therefore, most patients turn to surgical intervention. In previous studies, percutaneous vertebroplasty (PVP) and percutaneous kyphoplasty (PKP) were reported to provide good results in the treatment of stage I and II Kümmell’s disease due to their less invasive, shorter operation time, and significant improvement in the patient’s quality of life [4].

Percutaneous vertebral augmentation (PVA) has achieved satisfactory results in pain relief and deformity correction [5]. However, both PVP and PKP may cause a catastrophic complication of intraoperative or postoperative bone cement displacement due to the untight combination of bone cement cancellous bone [6,7]. Some scholars believed that the displacement of bone cement might cause vertebral collapse and spinal instability, resulting in persistent pain, kyphosis aggravation, and neurological damage [7,8,9,10].

Up to now, bone cement displacement following PVA is influenced by unclear factors, and systematic studies are lacking. In addition, most of the literature about bone cement displacement are case reports, and the related factors are not described in detail. Therefore, a case-control study to analyze the clinical data of 824 patients who underwent PVP or PKP therapy for stages I and II of Kümmell’s disease in Honghui Hospital affiliated with Xi’an Jiaotong University from January 2016 to June 2022 was conducted to explore the independent influencing factors of bone cement displacement.

## 2. Data and Methods

### 2.1. General Data

Inclusion criteria: (1) Patients with single-segment Kümmell’s disease. (2) Computed tomography (CT) examination confirmed the existence of intravertebral vacuum cleft (IVC) and intact posterior wall. (3) In Magnetic resonance imaging (MRI), low signal intensity was found on T1-weighted images, and either low or high signal intensity was found on T2-weighted images, depending on whether the gas or fluid fills the cleft. (4) The diagnoses of these patients were stage I or II Kümmell’s disease without neurological deficits [11]. (5) BMD T score less than −2.5 SD. (6) PVP or PKP was conducted.

Exclusive criteria: (1) Pathological fractures. (2) Patients with neurological deficits. (3) Multilevel Kümmell’s lesions. (4) Patients with infection at the puncture site. (4) Patients with severe cardiopulmonary dysfunction cannot undergo surgery. (5) Incomplete clinical data.

According to the imaging results of new back pain, the patients were divided into two groups: the bone cement displacement group (*n* = 150) and the non-cement displacement group (*n* = 674). The study was approved by the Medical Ethics Committee of Honghui Hospital affiliated with Xi’an Jiaotong University (20220311), and complied with the Helsinki Declaration’s ethical standards. Written informed consent was provided by all participants.

### 2.2. Bone Cement Displacement Diagnostic Criteria

The X-ray film showed a rupture of the anterior cortex of the vertebral body and anterior displacement of bone cement (Figure 1). CT showed a rupture of the anterior cortex of the vertebral body, the anterior edge of the bone cement was more than 2 mm from the anterior edge of the vertebral body, and the bone cement moved forward. MRI examination showed vertebral collapse, and sagittal T1-weighted images and T2-weighted images of the fracture cavity showed abnormally low and high signal intensity, respectively [12]. A typical case is shown in Figure 2.

### 2.3. Treatment Method

All procedures were performed by senior physicians with PVA qualifications. The patient was in a prone position, and fluoroscopic positioning was performed by a C-arm X-ray machine. Local anesthesia was performed on the articular capsule, the local muscle, and the fascia before diaplasis (0.5% lidocaine). The needle was inserted through unilateral or bilateral pedicle surface projection positioning of the surgical vertebrae. Then the guide needle, dilation cannula, and work cannula were replaced sequentially under fluoroscopy. The C-arm X-ray machine fluoroscopy determined that the puncture needle exceeded the posterior edge of the vertebral body, and the needle tip did not cross the medial cortex. The needle core was pulled out after the satisfactory position was determined by lateral fluoroscopy. Surgical procedures for PVP patients: under the fluoroscopy of the C-arm X-ray machine, the pressure syringe was used to inject the high-viscosity or low-viscosity bone cement into the responsible vertebra. Surgical procedures for PKP patients: prior to injection of bone cement, a deflated balloon was placed into the vertebral body and inflated to restore the height of surgical vertebrae and the balloon was then deflated and withdrawn. In addition to using the balloon, the operation steps of PKP were similar to the previous method. Finally, the diffusion, distribution, displacement, and leakage of bone cement were observed under the fluoroscopy of the C-arm X-ray machine. After satisfactory, the injection device was removed, the wound was compressed to stop bleeding, and a sterile dressing was applied. After the operation, the patients were transferred to the observation room for observation. After the vital signs were normal, they returned to the ward and performed rehabilitation exercises under the protection of the brace. All patients were instructed to exercise and ordered to receive standardized anti-osteoporosis treatment and regular review.

### 2.4. Evaluation Index

The following patient data were collected: gender, age, BMI, underlying disease, BMD, surgical method, surgical approach, the volume of cement, the viscosity of cement, cement leakage, postoperative anti-osteoporosis treatment, and postoperative bracing. Radiologic images recorded involved vertebral segment, Kümmell’s disease staging, the integrity of anterior vertebral cortex, the integrity of endplate in surgical vertebrae, the preoperative and post-operative anterior height of surgical vertebrae, restoration of anterior height of vertebra, pre-operative and post-operative Cobb angles, local restoration of Cobb angle, distribution of cement, and cement leakage.

### 2.5. Index Definition

BMI (kg/m^2^) was calculated as weight (kg) divided by the height squared (m^2^). The thoracolumbar junction is defined as T11–L2. In terms of Kümmell’s disease staging, stage I: vertebral body height loss <20%, with or without adjacent intervertebral disc degeneration; and stage II: vertebral body height loss >20% along with adjacent disc degeneration. Cement leakage was defined by postoperative X-ray or CT examination showing that the bone cement exceeds the upper and lower endplates. The local Cobb angle was defined as the angle between the inferior endplate of the superior vertebra of the fractured vertebra and the superior endplate of the inferior vertebra of the fractured vertebra. The restoration rate of vertebral height was calculated as follows: (anterior height of fractured vertebra/([upper adjacent vertebral anterior height + lower adjacent vertebral anterior height]/2)). The local restoration of the Cobb angle was calculated as follows: ((preoperative Cobb angle—postoperative Cobb angle)/preoperative Cobb angle) [13] (Figure 3). The even cement distribution is characterized by the spongiform dispersive bone cement in contact with the upper and lower endplates.

### 2.6. Statistical Analysis

All data were analyzed with SPSS version 26 (IBM, Armonk, NY, USA) and GraphPad Prism version 9.3.0 (San Diego, CA, USA). Quantitative data were presented as the mean ± standard deviation (SD) and compared with Student’s *t*-test, while categorical data were compared using Fisher’s exact test and chi-square test. The indexes with statistical differences were input into binary logistic regression correlation analysis to determine the risk factors of bone cement displacement. The discrimination ability was evaluated using the area under the curve (AUC) of the receiver operating characteristic (ROC). The predefined significance level for inclusion in the regression model was a *p*-value of 0.05.

## 3. Results

### 3.1. General Information

A total of 824 patients with stage I or II Kümmell’s disease were enrolled, including 248 males and 576 females with a mean age of 70 years (range 55–95 years). The study population consisted of 219 cases with stage Ⅰ Kümmell’s disease and 605 cases with stage Ⅱ Kümmell’s disease. BMD with T-value ranging from −2.5 to −5.4 SD (mean, −3.6 SD). Among the 824 vertebrae addressed, 619 (75.1%) involved the thoracolumbar junction (T11–L2), 121 (14.7%) involved the thoracic junction (T5–T10), and 84 (10.2%) involved the lumbar junction (L3–L5). The time of post-operative cement displacement ranges from 1.8 to 38.2 months (median, 15). The associated clinical and radiological characteristics are described in Table 1.

### 3.2. Univariate Analysis

The results of the univariate analysis showed that the bone cement displacement after PVA was not correlated with gender, age, BMI, underlying diseases, preoperative anterior vertebral height, postoperative anterior vertebral height, restoration of the anterior height of the vertebra, surgical approach cement volume, or postoperative bracing (*p* > 0.05). Involved vertebral segment, BMD, Kümmell’s disease staging, local restoration of the Cobb angle, the integrity of anterior vertebral cortex, the integrity of endplate in surgical vertebrae, surgical method, distribution of cement, the viscosity of cement, cement leakage, and postoperative anti-osteoporosis treatment were associated with bone cement displacement following PVA (*p* < 0.05, Table 2).

### 3.3. Binary Logistic Regression Analysis

In the logistic regression analysis, the risk factors of bone cement displacement included thoracolumbar junction (odds ratio (OR) = 3.23, 95% confidence interval (CI) 2.12–4.50, *p* = 0.011), Kümmell’s disease staging (OR = 2.23, 95% CI 1.81–3.41, *p* < 0.001), anterior cortex defect (OR = 5.34, 95% CI 3.53–7.21, *p* < 0.001), endplates defect (OR = 0.54, 95% CI 0.35–0.71, *p* < 0.001), cement distribution (OR = 2.86, 95% CI 2.03–3.52, *p* = 0.002), cement leakage (OR = 4.59, 95% CI 3.85–5.72, *p* < 0.001), restoration of the local Cobb angle (OR = 3.17, 95% CI 2.40–5.73, *p* = 0.024), and postoperative anti-osteoporosis treatment (OR = 0.48, 95% CI 0.18–0.72, *p* = 0.025) were independently associated with the bone cement displacement following PVA (Table 3 and Figure 4). The results of ROC curve analysis showed that the AUC of these independent influencing factors for predicting bone cement displacement after PVA in Kümmell’s disease was 0.816 (95% CI 0.747–0.885), the sensitivity was 0.717, and the specificity was 0.793 (Figure 5).

## 4. Discussion

Bone cement displacement is a rare complication after PVA for Kümmell’s disease. At present, the influencing factors of bone cement displacement have not been studied, and only a few cases have been reported [7,8,9,12]. In short, there is no systematic study on bone cement displacement after PVA in Kümmell’s disease. Therefore, combined with clinical practice, we deeply studied the influencing factors of bone cement displacement after PVA in Kümmell’s disease. The result of binary logistic regression analysis showed that in this study, thoracolumbar junction, Kümmell’s disease staging, anterior cortex defect, vertebral endplates defect, cement distribution, cement leakage, restoration of the local Cobb angle, and postoperative anti-osteoporosis treatment were independently associated with the bone cement displacement after PVA in Kümmell’s disease.

Consistent with previous results, we found that the thoracolumbar junction was an unignorable contributor to bone cement displacement. In this study, a higher rate of bone cement displacement was observed in patients whose initial fracture occurred in the thoracolumbar junction [14]. The thoracolumbar junction is located at the turning point of physiological kyphosis and kyphosis. It has a large range of motion and is the stress concentration area of the whole spine. The corresponding increase of thoracolumbar kyphosis and lumbar lordosis after the initial fracture will lead to a change in the center of gravity and the stress of the vertebral body, which is easy to cause the micromovement of bone cement in the vertebral body. Nagad et al. [10] believed that intravertebral instability was an important reason for low back pain symptoms and bone cement migration after PVP in Kümmell’s disease. Previous studies have shown that injection of bone cement into the thoracolumbar segment changes the elastic modulus of the vertebral body and the curvature of the spine, which undoubtedly increases the risk of cement fretting and causes delayed displacement of bone cement over time [15].

Li et al. [11] divided Kümmell’s disease into three stages based on the existence of adjacent intervertebral disc degeneration, vertebral height collapse and spinal cord compression. To be specific, the vertebral body height loss is less than 20% in stage Ⅰ of Kümmell’s disease, regardless of adjacent intervertebral disc degeneration. Stage II is characterized by a reduction in vertebral body height greater than 20% along with adjacent disc degeneration. Stage Ⅲ Kümmell’s disease is characterized by posterior breakage with or without spinal cord compression. In this study, the proportion of stage Ⅱ Kümmell’s disease in the cement displacement group was significantly higher than that in the non-displacement group. Logistic regression analysis showed that Kümmell’s disease staging was an independent risk factor for cement displacement. Therefore, the operator should review the imaging data carefully preoperatively and be careful with severe compression fractures intraoperatively.

Interestingly, the anterior cortex defect was an independent risk factor for cement displacement in this study, while the endplate defect was a protective factor. Wang et al. [8] suggested that anterior cortical incompleteness may increase the chance of anterior migration of the bone cement under weight bearing, which is consistent with our results. Tsai et al. [7] also believed that the integrity of the anterior cortex is an important consideration of anterior dislodgment. In addition, bone cement tends to be distributed at the anterior edge under the action of injection pressure in the PVA, but in the case of anterior cortex defect, bone cement is more likely to be dispersed to the anterior vertebral body. However, a channel is formed between the defect area of the endplates and the intervertebral disc space when the bone cortex ruptures on the upper and lower endplates, and the bone cement diffuses into the intervertebral space through this channel with little resistance. Bone cement forms root-like connections between vertebral endplates, forming a good chimeric relationship and reducing the occurrence of bone cement displacement. In addition, pathology confirmed that there was obvious fibroproliferative scar tissue or hardened necrotic bone on the surface of IVC, which was not conducive to fracture healing and cement-bone interface stability. Bone insufficient penetration of bone cement into trabeculae after cement filling may be another important reason for bone cement displacement after surgery.

The uneven bone cement distribution in surgical vertebrae is considered to be the main influencing factor of the poor postoperative effect [16]. The results of the study showed that the uneven distribution of bone cement is another consideration of cement displacement. When a vertebrae body is affected by trabecular ischemic necrosis, there will be an intraspinal fissure, which is weak around the cavity. The anchorage of the defect area is poor, and the pressure in the fissure is low. As a result, the cement is unable to disperse adequately and tends to form clumps, resulting in decreased strength and stability of the vertebral body. Liang et al. [17] showed through finite element analysis that bone cement distributed in clumpy mass would increase the stress of surrounding cancellous bone, which destroys the stability of the cement-bone interface. The cancellous bone between the cement and the endplate is prone to micromovement under long-term stress, resulting in bone cement displacement gradually. On the contrary, Gao et al. [12] believed that even bone cement distribution and contact with the upper and lower endplates of the vertebral body could significantly reduce the risk of postoperative cement displacement. When the bone cement is evenly distributed and in contact with the upper and lower endplates, the load on the vertebral body can be orderly and evenly transmitted according to the order of the upper endplate, the bone cement, and the lower endplate, which enhances the stiffness and strength of the vertebral body and significantly reduces the probability of bone cement micromotion in the vertebral body. Therefore, it is recommended to use a spoon scraper to remove the hardened bone around the vertebral fissure so as to make the bone cement disperse more fully in the vertebral body, increase the interweaving of bone cement and cancellous bone, and reduce the occurrence of bone cement displacement.

Bone cement leakage is a common complication after PVA, with an incidence from 11% to 73% [18]. The results of this study showed that bone cement leakage was significantly associated with the occurrence of bone cement displacement following PVA. The presence of bone cement leakage affects spinal biomechanics to a certain extent, and the risk of bone cement displacement increases over time. Kawaguchi et al. [19] found an increased risk of cement loosening after cement leakage during follow-up. Nagad et al. [10] also believed that cement leakage could cause intervertebral instability or intravertebral instability, resulting in the delayed displacement of bone cement. Therefore, when injecting bone cement, it should be under close fluoroscopy control and halted immediately if a spillover occurs. Avoiding cement leakage may reduce the risk of cement displacement.

The results of this study show that high restoration of the local Cobb angle is an independent risk factor for bone cement displacement following PVA in Kümmell’s disease, which is consistent with the results of a previous study [12]. Mao et al. [20] and Dai et al. [21] found that the restoration of the local Cobb angle was the influencing factor of vertebral re-fracture after PKP and PVP by meta-analysis. Kang et al. [22] found that the local kyphotic angle was significantly related to vertebral refracture and considered that the high restoration of the local Cobb angle was a risk factor. It is possible that the high restoration of the local Cobb angle is due to too small postoperative Cobb angles, which can lead to sagittal spinal instability, sagittal spinal stress imbalance, and increased bone cement-trabecular interface stress. With frequent activity and increased activity in the early postoperative period, the relative stability between bone cement and trabecular bone will be broken and delayed cement displacement will undoubtedly occur.

Osteoporosis is a chronic systemic metabolic bone disease [23]. Previous studies have shown that osteoporosis causes microstructural changes in the vertebral body and increases local stress, which will lead to the displacement of bone cement after surgery [2]. Effective antiosteoporosis drugs can correct the imbalance of bone turnover, maintain bone trabecular structure, improve bone mechanical strength, and improve the symptoms of osteoporosis [24]. This study showed that standardized anti-osteoporosis therapy was a protective factor for postoperative cement displacement. Ma et al. [25] also showed that elderly patients with osteoporotic vertebral compression fracture without standardized anti-osteoporosis postoperatively had a higher incidence of re-fracture. A meta-analysis of 9372 patients showed that anti-osteoporosis treatment was a protective factor for vertebral re-fracture after PVA [21]. In a 34-year prospective study, Svejme et al. [26] showed that low BMD was an independent risk factor for fracture and high mortality after menopausal. In clinical practice, adhering to standardized treatment is an important factor in the efficacy of anti-osteoporosis drugs. Therefore, physicians should do a good job of health education for patients during the perioperative period, clearly inform patients that they belong to the population with severe osteoporosis, take anti-osteoporosis drugs strictly according to the course of treatment, and pay close attention to the patients’ anti-osteoporosis treatment.

## 5. Limitation

This study is a retrospective single-center study with a small number of cases, and the data contained do not record the location and severity of bone cement leakage. A prospective, multicenter study with a larger sample size is needed to verify the conclusions of this study.

## 6. Conclusions

Thoracolumbar fracture, stage Ⅱ Kümmell’s disease, anterior cortex defect, uneven cement distribution, cement leakage, and the high restoration of the local Cobb angle are independent risk factors for bone cement displacement after PVA in Kummell’s disease. Vertebral endplates defect and postoperative anti-osteoporosis treatment were protective factors.

## Figures and Tables

**Figure 1 jcm-11-07479-f001:**
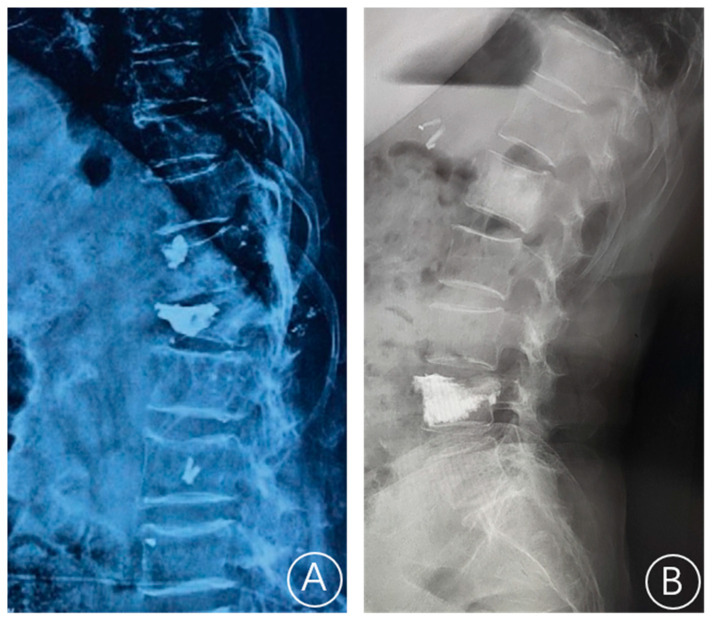
A plain radiograph showed whether the bone cement is displaced. (**A**) Bone cement displacement. (**B**) Non-bone cement displacement.

**Figure 2 jcm-11-07479-f002:**
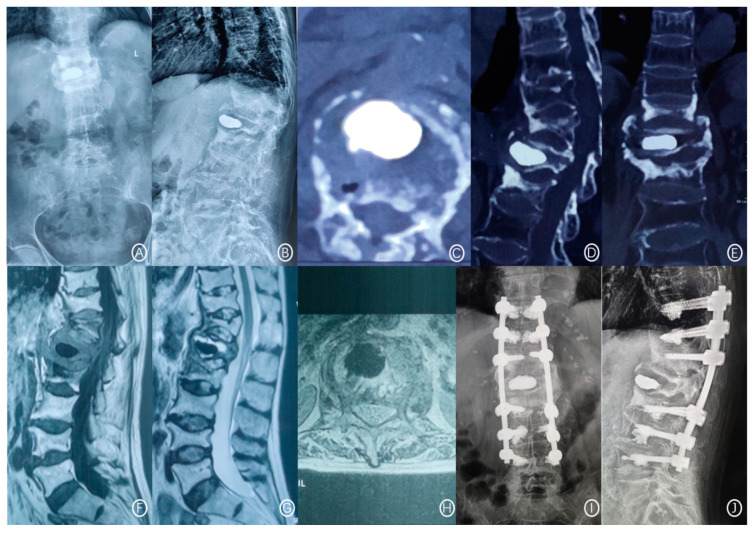
A 71-year-old female patient had recurrent low back pain and limited movement 3 years after L_1_ vertebroplasty. (**A**,**B**): Lumbar positive and lateral X-ray films scanning at 36 months after PVP showed L_1_ vertebral cement displacement and T_12_ vertebral compression fracture. (**C**–**E**): CT cross-section, sagittal plane, and coronal plane showed anterior cortex defect and bone cement displacement of the L_1_ vertebral body and compression fracture of the T_12_ vertebral body. (**F**–**H**): T_1_W_1_ showed low signal intensity, and T_2_W_1_ showed high signal intensity at the L_1_ vertebral body. (**I**,**J**): The positive and lateral positions of X-ray films showed that after posterior thoracolumbar bone graft fusion and bone cement enhanced internal fixation.

**Figure 3 jcm-11-07479-f003:**
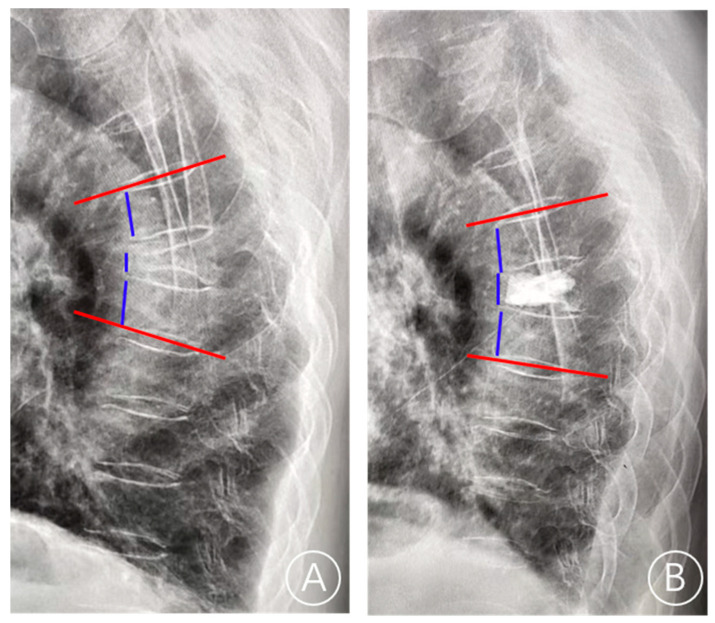
The lateral radiograph showed the measurement of the anterior vertebral height and the local Cobb angle. (**A**) On the preoperative lateral radiographs, the anterior height of the fractured vertebra, the upper adjacent vertebral anterior height, the lower adjacent vertebral anterior height, and the Cobb angle were measured. (**B**) On the postoperative lateral radiographs, the anterior height of the fractured vertebra, the upper adjacent vertebral anterior height, the lower adjacent vertebral anterior height, and the Cobb angle were measured. The red line is the endplate and the blue line is the anterior vertebral height.

**Figure 4 jcm-11-07479-f004:**
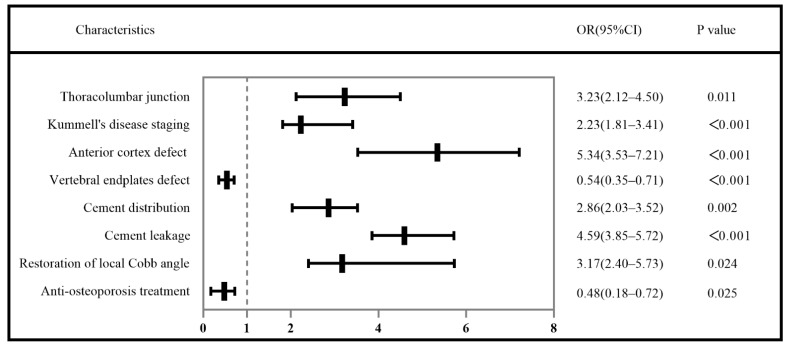
Risk factors of postoperative bone cement displacement in Kümmell’s disease.

**Figure 5 jcm-11-07479-f005:**
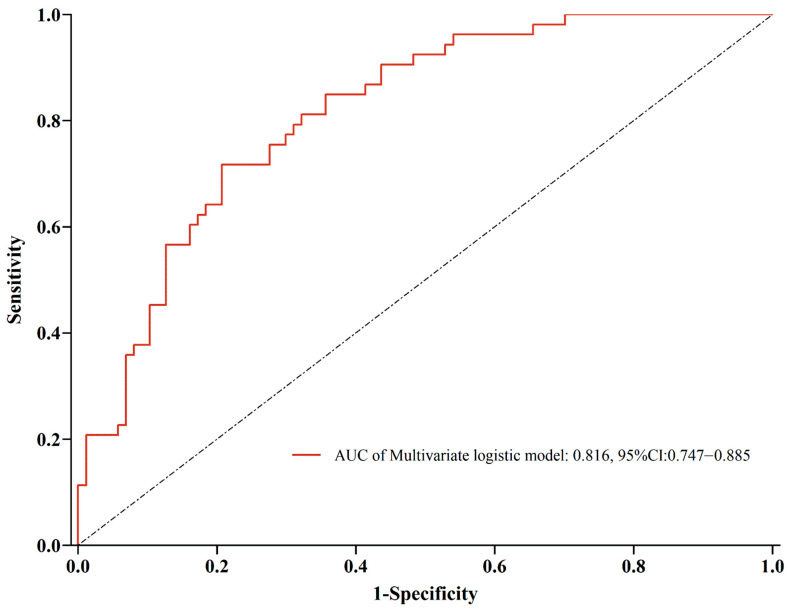
Receiver operating curves (ROC) for the prediction performance of the logistic regression model.

**Table 1 jcm-11-07479-t001:** Characteristics of patient group.

Number of Patients, *n*	824
Gender	
Male/Female	248/576
Mean age (range, years)	70 (55–95)
Mean BMI (range, kg/m²)	24.0 (19.1–28.0)
Underlying diseases	
Hypertension/Diabetes/Heart disease/Others	121/119/70/514
Mean BMD (range)	−3.6 (−2.5–−5.4)
Injured vertebral segment	
Thoracolumbar junction/thoracic junction/lumbar junction	619/121/84
Kümmell’s disease staging	
I/II	219/605
Median cement displacement (range, months)	15 (1.8–38.2)
Mean preoperative anterior vertebral height (range, mm)	25.0 (16.3–29.5)
Mean postoperative anterior vertebral height (range, mm)	28.0 (23.1–31.6)
Mean preoperative local Cobb angles (range, °)	22.8 (11.8–28.3)
Mean postoperative local Cobb angles (range, °)	13.1 (9.1–16.7)
Integrity of anterior vertebral cortex	
No/Yes	406/418
Integrity of endplate in surgical vertebrae	
No/Yes	150/674
Surgical method	
PVP/PKP	189/635
Surgical approach	
Unilateral/Bilateral	678/146
Mean volume of cement (range, mL)	4.5 (2.4–6.7)
Even cement distribution	
No/Yes	365/459
Viscosity of cement	
High/Low	531/293
Cement leakage	
No/Yes	405/419
Postoperative anti-osteoporosis treatment	
No/Yes	511/313
Postoperative bracing treatment	
No/Yes	239/585

Abbreviations: BMI—body mass index; BMD—bone mineral density; PVP—percutaneous vertebroplasty; and PKP—percutaneous kyphoplasty.

**Table 2 jcm-11-07479-t002:** Univariate analysis of postoperative bone cement displacement following PVA in Kümmell’s disease.

Characteristic	Displacement Group (*n* = 150)	Non-Displacement Group (*n* = 674)	*χ*^2^/*t* Value	*p*-Value
Gender			3.04	0.081
Male	54	194		
Female	96	480		
Age (years, $\bar{x}$ ± *s*)	73.9 ± 7.1	75.1 ± 6.9	1.92	0.06
BMI (kg/m^2^, $\bar{x}$ ± *s*)	24.2 ± 2.1	23.9 ± 2.9	1.47	0.144
Underlying diseases			6.55	0.088
Hypertension	26	95		
Diabetes	30	89		
Heart disease	11	59		
Others	83	431		
BMD ($\bar{x}$ ± *s*)	−3.8 ± 0.9	−3.5 ± 1.9	2.89	0.004
Thoracolumbar junction			10.72	0.001
No	53	152		
Yes	97	522		
Kümmell’s disease staging			8.34	0.004
I	54	165		
II	96	509		
Preoperative anterior vertebral height (mm)	25.30 ± 3.51	25.62 ± 4.11	0.98	0.329
Postoperative anterior vertebral height (mm)	26.84 ± 3.08	27.02 ± 4.02	0.61	0.543
Restoration of anterior height of vertebra (%)	2.50 ± 0.11	2.52 ± 0.23	1.59	0.114
Preoperative local Cobb angles (°)	22.96 ± 4.72	23.79 ± 4.91	1.89	0.060
Postoperative local Cobb angles (°)	12.87 ± 3.89	13.11 ± 3.33	0.70	0.484
Restoration of local Cobb angle (%)	41.04 ± 7.81	32.67 ± 8,52	11.38	<0.001
Integrity of anterior vertebral cortex			14.51	<0.001
No	95	311		
Yes	55	363		
Integrity of endplate in surgical vertebrae			9.66	0.002
No	62	373		
Yes	88	301		
Surgical method			4.99	0.026
PVP	24	165		
PKP	126	509		
Surgical approach			2.31	0.129
Unilateral	117	561		
Bilateral	33	113		
Volume of cement (mL)	4.54 ± 1.12	4.72 ± 1.56	1.65	0.101
Even cement distribution			16.81	<0.001
No	89	276		
Yes	61	398		
Viscosity of cement			4.57	0.033
High	108	423		
Low	42	251		
Cement leakage			7.07	0.008
No	59	346		
Yes	91	328		
Postoperative anti-osteoporosis treatment			11.18	<0.001
No	111	400		
Yes	39	274		
Postoperative bracing treatment			2.85	0.091
No	52	187		
Yes	98	487		

Abbreviations: BMI—body mass index; BMD—bone mineral density; PVP—percutaneous vertebroplasty; and PKP—percutaneous kyphoplasty.

**Table 3 jcm-11-07479-t003:** Binary logistic regression correlation analysis for risk factors of bone cement displacement following PVA in Kümmell’s disease.

Characteristic	OR Value	95% CI	*p* Value
BMD	3.56	0.79–4.12	0.109
Thoracolumbar junction	3.23	2.12–4.50	0.011
Kümmell’s disease staging	2.23	1.81–3.41	<0.001
Anterior cortex defect	5.34	3.53–7.21	<0.001
Vertebral endplates defect	0.54	0.35–0.71	<0.001
Surgical method	1.54	0.84–1.79	0.413
Cement distribution	2.86	2.03–3.52	0.002
Viscosity of cement	1.36	0.78–1.77	0.154
Cement leakage	4.59	3.85–5.72	<0.001
Restoration of local Cobb angle	3.17	2.40–5.73	0.024
Postoperative anti-osteoporosis treatment	0.48	0.18–0.72	0.025

Abbreviations: BMD—bone mineral density.

## Data Availability

The data presented in this study are available on request from the corresponding author. The data are not publicly available due to privacy restrictions.

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
