# Peer review of "Predictive Factors for Bone Cement Displacement following Percutaneous Vertebral Augmentation in Kümmell’s Disease"

_jcm, 2022, doi:10.3390/jcm11247479_

Round 1
Reviewer 1 Report
Thankyou for giving me the possibility to review this interesting paper aiming to investigate the independent influencing factors of bone cement displacement following percutaneous vertebral augmentation in patients with stage I and stage II Kümmell's disease. The paper is well written and the results are supported by a robust statistical analysis. Limitations have been clearly reported as well.
However, before considering it for pubblication, the following concerns should be addressed:
1. Please detail the acute treatment of the vertebral compression fractures (have they been diagnosed or the diagnosis was missed? How have they beeen managed?
2. Please add the prevalence of KD in your hospiral in the study period
3. Please add a gender specific analysis in the results section
4. Please detail how many patients have a diagnosis of osteoporosis
5. Have you performed DEXA scan in all the recruited patients? If so, could you provide the mean femoral and lumbar T-scores?
6. Please state if you assessed vit D concentration and correction
Author Response
Comment 1: Thank you for giving me the possibility to review this interesting paper aiming to investigate the independent influencing factors of bone cement displacement following percutaneous vertebral augmentation in patients with stage I and stage II Kümmell's disease. The paper is well written and the results are supported by a robust statistical analysis. Limitations have been clearly reported as well. However, before considering it for publication, the following concerns should be addressed:
Response: We would like to thank you for your careful reading, helpful comments, and constructive suggestions, which have significantly improved the presentation of our manuscript. We have carefully considered all comments from the reviewers and revised our manuscript accordingly. We hope our revised manuscript can be accepted for publication.
Comment 2: Please detail the acute treatment of the vertebral compression fractures (have they been diagnosed or the diagnosis was missed? How have they been managed?
Response: Osteoporotic vertebral fracture is a low-energy injury. In addition, the elderly patients have a slow response to pain, which does not attract enough attention at first, often leading to delays in the condition and missing the best opportunity for treatment. Fresh fracture transforms into Kümmell's disease, which leads to prolonged low back pain and seriously affects the quality of life of patients. Treatment for the acute vertebral compression fractures includes conservative and surgical approaches. In acute stages, the fractures are well treated with conservative measures including short bed rest, analgesics, bracing, and exercises. Studies have shown that although most fractures heal well, up to 30% of fractures can develop painful nonunion, progressive kyphosis, and neurological deficit. For patients who develop severe pain not responding to nonoperative measures and painful nonunion, percutaneous cement augmentation procedures including vertebroplasty or kyphoplasty have been suggested.
All the cases included in our study were complicated with vertebral fracture on the basis of osteoporosis, which developed into vertebral collapse, vertebral fissure sign, vertebral fracture nonunion and vertebral pseudarthrosis after more than 3 months of standard treatment or without standard treatment. CT examination confirmed the existence of intravertebral vacuum cleft (IVC) and intact posterior wall. In MRI, low signal intensity was found on T1-weighted images, and either low or high signal intensity was found on T2-weighted images, depending on whether the gas or fluid fills the cleft.
Reference:
- Rajasekaran S, et al. Osteoporotic Thoracolumbar Fractures-How Are They Different?-Classification and Treatment Algorithm. J Orthop Trauma. 2017 Sep;31 Suppl 4:S49-S56.
- Alsoof D, et al. Diagnosis and Management of Vertebral Compression Fracture. Am J Med. 2022 Jul;135(7):815-821.
Comment 3: Please add the prevalence of KD in your hospital in the study period
Response: The prevalence of KD in our hospital from 2016 to June 2022 was approximately 0.167 per 1000 population.
Comment 4: Please add a gender specific analysis in the results section
Response: Thank you for your comments. The results of univariate analysis showed that the bone cement displacement after PVA was not correlated with gender.
Comment 5: Please detail how many patients have a diagnosis of osteoporosis
Response: Thank you for your comments. All the recruited patients were diagnosed with osteoporosis by Dual-energy X-ray absorptiometry. The diagnostic criteria for osteoporosis is T score ≤-2.5 SD.
Comment 6: Have you performed DEXA scan in all the recruited patients? If so, could you provide the mean femoral and lumbar T-scores?
Response: Yes. Unfortunately, we only collected the mean femoral T-score. Because we considered degenerative changes of the lumbar spine include lumbar facet joint hyperplasia, calcification of anterior and posterior longitudinal ligament, ligamentum flavum and abdominal aorta, and vertebral bone hyperplasia, which might affect the actual bone mineral density.
Comment 7: Please state if you assessed vit D concentration and correction
Response: Thank you for your comments. We did not assess vit D concentrations and corrections. All patients were required to receive standard anti-osteoporosis therapy in the department of osteoporosis, and the concentration and correction were not available.
Reviewer 2 Report
Large study with some interesting insights. Some questions and comments to address:
- Did some of these patients have to wear a brace after surgery? was this a predictive factor?
- Please clearly define what you mean by cement displacement? Do you mean new bone fracture? Do you mean cement leaking? This is not clear to me
- Please include definitions and figures of type 1 and 2 kummels disease, kummels disease is rarely categorized and it would be great to see the difference.
- Am I correct in assuming that Percuteanous vertebral augementation is just cement through perforated screws, and that percuteanous vertebroplasty is just transpendicular cement placement? please clarify
- what was the indication for these procedures, please calrify.
- please standardize the font which changes multiple types between texts in the document
Author Response
Comment 1: Large study with some interesting insights. Some questions and comments to address:
Response: We would like to thank you for your careful reading, helpful comments, and constructive suggestions, which have significantly improved the presentation of our manuscript. We have carefully considered all comments from the reviewers and revised our manuscript accordingly. We hope our revised manuscript can be accepted for publication.
Comment 2: Did some of these patients have to wear a brace after surgery? was this a predictive factor?
Response: Thank you for your valuable comments. We have added this part to the text and the table. But the result of univariate analysis showed that the bone cement displacement was not correlated with postoperative bracing treatment (P>0.05). In addition, previous studies have shown that bracing after percutaneous vertebroplasty for thoracolumbar osteoporotic vertebral compression fractures was not effective.
Reference: Zhang J, et al. Bracing after percutaneous vertebroplasty for thoracolumbar osteoporotic vertebral compression fractures was not effective. Clin Interv Aging. 2019 Feb 5; 14:265-270.
Comment 3: Please clearly define what you mean by cement displacement? Do you mean new bone fracture? Do you mean cement leaking? This is not clear to me
Response: Thank you for your kind comments. Cement displacement is neither refracture nor cement leakage, which is a catastrophic complication during or after surgery due to poor binding of cement to cancellous bone. Its diagnostic criteria included: the X-ray film showed rupture of the anterior cortex of the vertebral body and anterior displacement of bone cement. CT showed rupture of the anterior cortex of the vertebral body, the anterior edge of the bone cement was more than 2mm from the anterior edge of the vertebral body, and the bone cement moved forward. MRI examination showed vertebral collapse, and sagittal T1-weighted images and T2-weighted images of fracture cavity showed abnormally low and high signal intensity, respectively.
Reference:
- Jeong, Y.H, et al.Insufficient Penetration of Bone Cement Into the Trabecular Bone: A Potential Risk for Delayed Bone Cement Displacement After Kyphoplasty? Reg Anesth Pain Med 2016, 41 (5), 616-618.
- Tsai, T.T, et al. Polymethylmethacrylate cement dislodgment following percutaneous vertebroplasty: a case report. Spine (Phila Pa 1976) 2003, 28 (22), E457-460.
- Gao X, et al. Risk factors for bone cement displacement after percutaneous vertebral augmentation for osteoporotic vertebral compression fractures. Front Surg. 2022 Jul 28; 9:947212.
Comment 4: Please include definitions and figures of type 1 and 2 kummels disease, kummels disease is rarely categorized and it would be great to see the difference.
Response: Thank you for your comments. The definitions type 1 and 2 Kümmell’s disease was added in the Index definition. In terms of Kümmell's disease staging, stage I: vertebral body height loss <20%, with or without adjacent intervertebral disc degeneration; stage II: vertebral body height loss >20% along with adjacent disc degeneration.
Reference: Li KC, et al. Staging of Kümmell's disease. Journal of Musculoskeletal Research 2004, 08 (01), 43-55.
Comment 5: Am I correct in assuming that percutaneous vertebral augmentation is just cement through perforated screws, and that percutaneous vertebroplasty is just transpendicular cement placement? please clarify
Response: Thank you for your comments. Percutaneous vertebral augmentation includes percutaneous vertebroplasty (PVP) and percutaneous kyphoplasty (PKP), which is minimally invasive radiologically guided procedures that involve injection of polymethylmethacrylate (PMMA) bone cement into the vertebral body with the objective of achieving pain relief and preventing further loss of vertebral body heigh.
Comment 6: what was the indication for these procedures, please clarify.
Response: We treat these patients according to the latest guidelines. PVP and PKP have been suggested to treat stage I and II Kümmell's disease.
Reference:
- 陈伯华, et al.症状性陈旧性胸腰椎骨质疏松性骨折手术治疗临床指南.中华创伤杂志36.07(2020):577-586.
- Ito Y, et al. Pathogenesis and diagnosis of delayed vertebral collapse resulting from osteoporotic spinal fracture. Spine J. 2002;2(2):101-6.
Comment 7: please standardize the font which changes multiple types between texts in the document
Response: Thank you very much for your careful review. We have changed them according your comments.
Round 2
Reviewer 1 Report
The paper has been improved in this revised version. It could be accepted in the present form